# Cavity frequency-dependent theory for vibrational polariton chemistry

Xinyang Li[1], Arkajit Mandal [1✉] & Pengfei Huo [1,2✉]

Recent experiments demonstrate the control of chemical reactivities by coupling molecules inside an optical microcavity. In contrast, transition state theory predicts no change of the reaction barrier height during this process. Here, we present a theoretical explanation of the cavity modification of the ground state reactivity in the vibrational strong coupling (VSC) regime in polariton chemistry. Our theoretical results suggest that the VSC kinetics modification is originated from the non-Markovian dynamics of the cavity radiation mode that couples to the molecule, leading to the dynamical caging effect of the reaction coordinate and the suppression of reaction rate constant for a specific range of photon frequency close to the barrier frequency. We use a simple analytical non-Markovian rate theory to describe a single molecular system coupled to a cavity mode. We demonstrate the accuracy of the rate theory by performing direct numerical calculations of the transmission coefficients with the same model of the molecule-cavity hybrid system. Our simulations and analytical theory provide a plausible explanation of the photon frequency dependent modification of the chemical reactivities in the VSC polariton chemistry.

[1] Department of Chemistry, University of Rochester, Rochester, NY, USA. [2] The Institute of Optics, University of Rochester, Rochester, NY, USA.
✉email: amandal4@ur.rochester.edu; pengfei.huo@rochester.edu

Polariton Chemistry is an emerging field[1–5] that provides opportunities for new chemical reactivities or selectivities by coupling molecular systems to quantized radiation fields inside an optical cavity. By hybridizing electronic excitation of the molecule and the photonic excitation of the radiation inside the cavity, new light-matter entangled states, so-called polariton states are generated. Recent experimental and theoretical works have demonstrated the possibility of changing photo-isomerization reactivities[3,6–9], modifying electron transfer kinetics[10–12], and remotely controlling chemical reactions[13]. These new polaritonic photochemical reactivities are attributed to the modification of the excited state landscape[1,3,6–9,11,12] due to the formation of the polariton states.

Similarly, hybridizing molecular vibrations and the photonic excitations inside an optical cavity[14,15] forms vibrational polaritons (Fig. 1a). For the vibrational polaritonic hybrid system, it is a well-known result that the Rabi splitting observed in the infrared (IR) spectrum (due to light-matter couplings) scales as $\sqrt{N}$ with $N$ as the number of molecules[14,15] inside the cavity. Whether or not such a collective effect also manifests itself into chemical kinetics has been a subject of a debate[16–19]. Recent experiments have demonstrated that it is possible to suppress[20–24] or enhance[25,26] the ground-state chemical reactivities by placing an ensemble of molecules in an optical microcavity through the resonant coupling between the cavity and vibrational degrees of freedom (DOF) of the molecules. This so-called vibrational strong coupling (VSC) regime[5] operates in the absence of any light source[21,22], and was hypothesized to utilize the hybridization of a vibrational transition of a molecule and the zero-point energy fluctuations of a cavity mode[21,22]. This new strategy of VSC, if feasible, will allow one to bypass some intrinsic difficulties (such as intramolecular vibrational energy transfer) encountered in the mode-selective chemistry that uses IR excitation to tune chemical reactivities, offering a paradigm-shift of synthetic chemistry through cavity enabled bond-selective chemical transformations[21,22].

Unfortunately, a clear theoretical explanation of such remarkable VSC ground-state reactivities remains missing, including explaining both (i) the collective ($N$-dependent) effects on chemical reaction rates, and (ii) the resonant effect where the suppression of the rate is achieved with a particular cavity photon frequency. Recent theoretical works that use simple transition state theory (TST) suggest that there is no collective effect nor resonant effect in VSC polariton chemistry[17–19,27]. On the other hand, both effects do show up in a VSC non-adiabatic electron transfer reaction[28], with an enhancement of the rate upon resonant coupling between molecular vibration and the cavity, although the applicability of this theory on the VSC ground-state adiabatic reactions remains an open question.

In this work, we provide a different perspective on understanding the resonant effect of the VSC ground-state reactivities. Note that we refer to the photon frequency-dependent modification of the ground-state kinetics as the resonant effect. Through both analytical theory and numerical simulations, we demonstrate that the non-Markovian nature of a cavity radiation mode leads to significant suppression of the chemical reaction rate constant at a particular photon frequency that is related to the reaction barrier frequency. At such a "resonant" frequency, the cavity radiation mode induces the dynamical caging effect[29,30], such that the molecular reaction coordinate becomes trapped in a narrow "photonic solvent cage" near to the top of the barrier region, leading to a suppression of the chemical kinetics. Such effects are dynamical and are not captured within a simple transition state theory. This work underscores the importance of "dynamical solvent effect" of the cavity radiation modes and provides an understanding of the VSC polariton chemistry, paving the way toward an ultimate theoretical understanding of VSC polariton chemistry.

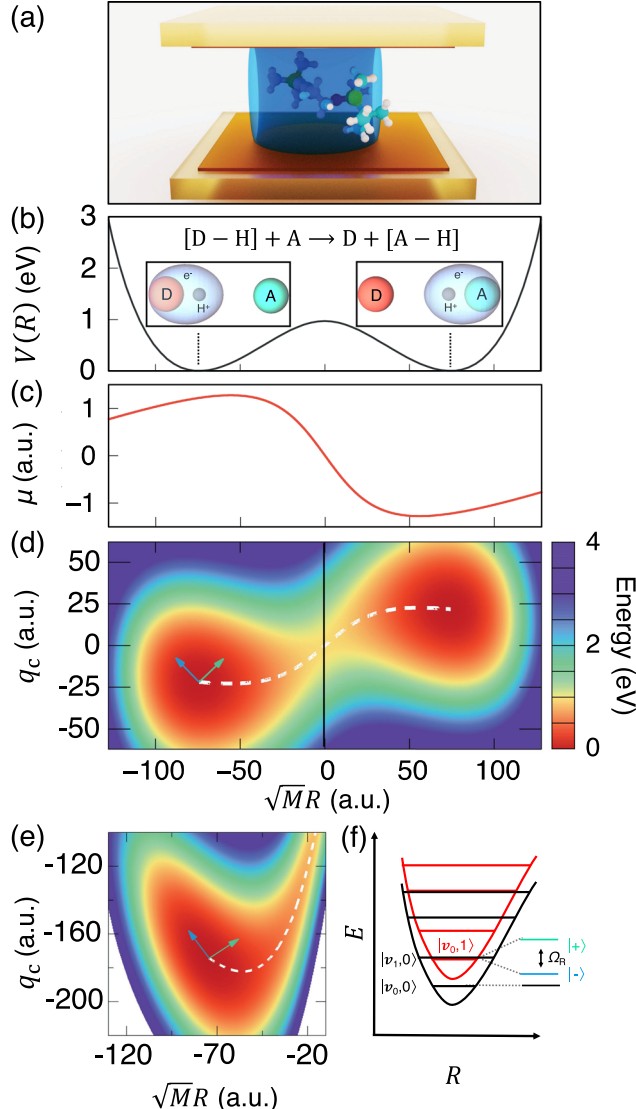

**Fig. 1 Vibrational strong coupling (VSC) regime in polariton chemistry.**
**a** Schematic representation of a molecule placed inside an optical cavity. **b** Proton-coupled electron transfer reaction of a Shin–Metiu model. Ground-state potential energy surface (PES) of the molecule as a function of the mass-weighted proton coordinate $\sqrt{M}R$ (in atomic unit) for the Shin–Metiu molecular model system. The ground-state electronic density at two different nuclear configurations (at the donor and acceptor minima) are illustrated in the insets. **c** Ground-state permanent dipole (solid red line) as a function of the mass-weighted proton coordinate $\sqrt{M}R$. **d** Cavity Born–Oppenheimer (CBO) surface along photonic coordinates $q_c$ and mass-weighted reaction coordinate $\sqrt{M}R$, with the white dash line representing the minimum energy path at the resonant frequency $\hbar\omega_0 = \hbar\omega_c = 0.1706$ eV and with a coupling strength $\eta = 0.047$. **e** A zoom-in to the reactant well of the CBO surface at the resonant frequency $\hbar\omega_c = 0.1706$ eV and $\eta = 0.376$. The arrows in **d** and **e** represent the directions of two polariton normal modes. **f** Schematic diagram showing the Rabi splitting $\hbar\Omega_R$ due to the light-matter coupling between photon-dressed vibronic-Fock states, $|\nu_0, 1\rangle$ (photonic excitation) and $|\nu_1, 0\rangle$ (vibrational excitation).

## Results

**Theoretical model.** The model QED Hamiltonian used in this work is expressed as[31–33]

$$\hat{H} = \frac{\hat{P}^2}{2M} + E(R) + \hat{H}_{\text{vib}} + \frac{\hat{p}_c^2}{2} + \frac{1}{2}\omega_c^2\left(\hat{q}_c + \sqrt{\frac{2}{\hbar\omega_c^3}}\chi \cdot \mu(R)\right)^2, \quad (1)$$

which is the Pauli–Fierz (PF) QED Hamiltonian (see "Methods") with the matter Hamiltonian operator and the dipole operator projected on the electronic ground state $|\Psi_g(R)\rangle$. Here, $E(R)$ is the ground-state potential energy surface for a Shin–Metiu (SM) model (an electron and a proton confined between two fixed charged ions) depicted in Fig. 1b, where $R$ is a proton transfer coordinate, $\mu(R) = \langle\Psi_g(R)|\hat{\mu}|\Psi_g(R)\rangle$ is the ground-state permanent dipole moment depicted in Fig. 1c, with $\hat{\mu}$ as the total dipole operator of the molecule. In addition, $\hat{H}_{vib}$ (see "Methods" for its expression) is the vibrational system-bath Hamiltonian that describes the interactions between reaction coordinate $R$ and other vibrational phonon modes in the molecule. Further, $\hat{q}_c = \sqrt{\hbar/2\omega_c}(\hat{a}^\dagger + \hat{a})$ and $\hat{p}_c = i\sqrt{\hbar\omega_c/2}(\hat{a}^\dagger - \hat{a})$ are the photon mode coordinate and momentum operator, respectively, where $\hat{a}^\dagger$ and $\hat{a}$ are the photon mode creation and annihilation operators. Under the dipole gauge, the matter interacts with the quantized radiation mode of the cavity by displacing the photonic coordinate (Fig. 1d–e) with the amount of $\sqrt{\frac{2}{\hbar\omega_c^3}}\chi \cdot \mu(R)$, where $\chi$ characterizes the coupling strength between the molecule and the cavity (see "Methods"). Note that the molecule-cavity coupling strength per molecule used in this work would be much stronger than the realistic coupling strength in the VSC experiments[21] that includes many molecules. On the other hand, the Rabi splitting (from the IR spectrum) of the current work is within the range of the recent VSC experiments[21,22]. This is because in these VSC experiments, the collective coupling strength is scaled up by $\sqrt{N}$. In this study, we have also explicitly assumed that the dipole moment is always aligned with the cavity polarization direction.

**Vibrational polariton Rabi splitting**. At the equilibrium position of the reactant $R_0$, one can approximate the permanent dipole as $\mu(R) \approx \mu_0 + \mu_0'(R - R_0)$, where $\mu_0 = \mu(R_0)$ and $\mu_0' = \frac{\partial\mu(R)}{\partial R}|_{R_0}$. The light-matter interaction term in $\hat{H}$ (Eq. (1)) at $R_0$ becomes[15,17] $\sqrt{\frac{2\omega_c}{\hbar}}\hat{q}_c\chi \cdot \mu(R_0) = \sqrt{\frac{\hbar}{2M\omega_0}}\chi \cdot \mu_0'(\hat{a}^\dagger + \hat{a})(\hat{b}^\dagger + \hat{b}) + \sqrt{\frac{2\omega_c}{\hbar}}\hat{q}_c\chi(\mu_0 - \mu_0'R_0)$, where $\omega_0 = \frac{\partial E^2(R)}{\partial R^2}|_{R_0}$ is the vibrational frequency at the equilibrium nuclear configuration $R_0$, $M$ is the effective mass of the nuclear vibration, $\hat{b}^\dagger$ and $\hat{b}$ are the creation and annihilation operators for the nuclear vibration associated with the coordinate $R$. At the resonant condition of $\omega_c = \omega_0$, the photon-vibration interaction couples photon-dressed vibronic-Fock states $|\nu_0, 1\rangle$ (photonic excitation) and $|\nu_1, 0\rangle$ (vibrational excitation), inducing a Rabi splitting $\hbar\Omega_R$ as follows[15,17]

$$\hbar\Omega_R = 2\sqrt{\frac{\hbar}{2M\omega_0}}\chi \cdot \mu_0' \equiv 2\hbar\omega_c \cdot \eta, \tag{2}$$

where the normalized coupling strength $\eta = \mu_0'\sqrt{\frac{\hbar}{2M\omega_0}}\frac{\chi}{\hbar\omega_c}$ characterizes the light-matter coupling strength. Note that the above relation between $\Omega_R$ and $\eta$ only holds under the linear approximation of the dipole operator, and it breaks down for ultra-strong coupling (USC) regime when $0.1 < \eta < 1$[34]. The $\hbar\Omega_R$ presented in Fig. 2 are instead obtained numerically from $\hat{H}$ (Eq. (1)), with details provided in Supplementary Fig. 4.

**Reaction rate constant**. The VSC polariton chemical kinetics can be viewed as a barrier crossing process on the cavity Born–Oppenheimer surface (CBO)[17,31,35] $V_{CBO}(R, q_c) = E(R) + \frac{1}{2}\omega_c^2(q_c + \sqrt{\frac{2}{\hbar\omega_c^3}}\chi \cdot \mu(R))^2$, which is a function of both $q_c$ and $R$. Note that the correct QED description in Eq. (1) includes the

dipole self-energy (DSE) $(\chi \cdot \mu(R))^2/\hbar\omega_c$ (see "Methods"). Without this term, one would get artificial changes of the barrier height and predicts a significant modification of the polariton potential energy barrier[17] (see Supplementary Fig. 3b). Since we are interested in the VSC regime, the cavity mode has a similar range of frequency as the molecular vibrations, meaning that $q_c$ evolve at a similar time scale as $R$. Based upon this consideration, we decide to follow the previous work[17–19] to treat both nuclear and photonic DOF classically. The electronic DOF is considered fully quantum mechanically, described by the adiabatic electronically ground-state wavefunction $|\Psi_g(R)\rangle$.

It is formally rigorous to express the rate constant as the TST rate $k_{TST}$ and the transmission coefficient $\kappa$ as follows

$$k = \lim_{t \to t_p} \kappa(t) \cdot k_{TST}, \tag{3}$$

where $t_p$ refers to the plateau time of the flux-side correlation function, and $\kappa(t)$ is the transmission coefficient that captures the dynamical recrossing effects, measuring the ratio between the reaction rate and the TST rate. It has been shown that classically the potential mean force is invariant to the change in coupling strength or photon frequency[18], and other theoretical investigations based on a simple TST analysis for $N$ molecules coupled to cavity also suggest no significant change of the reaction rate[19,27]. Since $k_{TST}$ does not change under the VSC condition, it is reasonable to conjecture that the change is purely dynamical and completely irrelevant to the potential barrier changes or free energy barrier changes. Thus, it is highly likely that VSC chemical reactivities are purely originated from the transmission coefficient $\kappa$. It can be numerically calculated from the flux-side correlation function formalism[36–38] as follows

$$\kappa(t) = \frac{\langle\mathcal{F}(0) \cdot h[R(t) - R_\ddagger]\rangle}{\langle\mathcal{F}(0) \cdot h[\dot{R}_\ddagger(0)]\rangle}, \tag{4}$$

where $h[R - R_\ddagger]$ is the Heaviside function of the reaction coordinate $R$, with the dividing surface $R_\ddagger$ that separate the reactant and the product regions (for the model system studied here, $R_\ddagger = 0$), the flux function $\mathcal{F}(t) = \dot{h}(t) = \delta[R(t) - R_\ddagger] \cdot \dot{R}(t)$ measures the reactive flux across the dividing surface (with $\delta(R)$ as the Dirac delta function), and $\langle\ldots\rangle$ represents the canonical ensemble average (subject to constrain on the dividing surface which is enforced by $\delta[R(t) - R_\ddagger]$ inside $\mathcal{F}(t)$). Further, $\dot{R}_\ddagger(0)$ represents the initial velocity of the nuclei on the dividing surface. The above flux-side formalism of the reaction rate can be derived from Onsager's regression hypothesis, with derivations presented in standard text books (e.g., ref.[38]). The numerical simulation details of $\kappa$ are provided in Supplementary Note 5.

To obtain a more intuitive understanding of how VSC light-matter interactions influence $\kappa$, let us consider a simplified model, $\hat{H} - \hat{H}_{vib}$ which only has two DOFs $\{R, q_c\}$ such that we can obtain an analytic expression of the rate as $k = k_{TST} \cdot \kappa_{GH}$. The transmission coefficient $\kappa_{GH}$ (under the limit $t \to t_p$) can be obtained from the Grote–Hynes (GH) theory[29,39–43]. The TST rate is $k_{TST} = \frac{\omega_0}{2\pi}e^{-\beta E_b}$, where $E_b = E(R_\ddagger) - E(R_0)$ is the potential energy barrier height measured from the bottom of the well $R_0$ to the top of the barrier $R_\ddagger$ (see Fig. 1b), and $\omega_0$ is the vibrational frequency of the reactant at $R = R_0$, and $\beta = (k_BT)^{-1}$. When explicitly considering the DSE, $E_b$ remains invariant as changing the light-matter coupling strength or the photon frequency (see Eq. (1)), explaining why one can not observe any effects from a simple TST analysis[18]. The total rate constant $k$ in the GH theory can be obtained using the multidimensional TST[40,44–46]

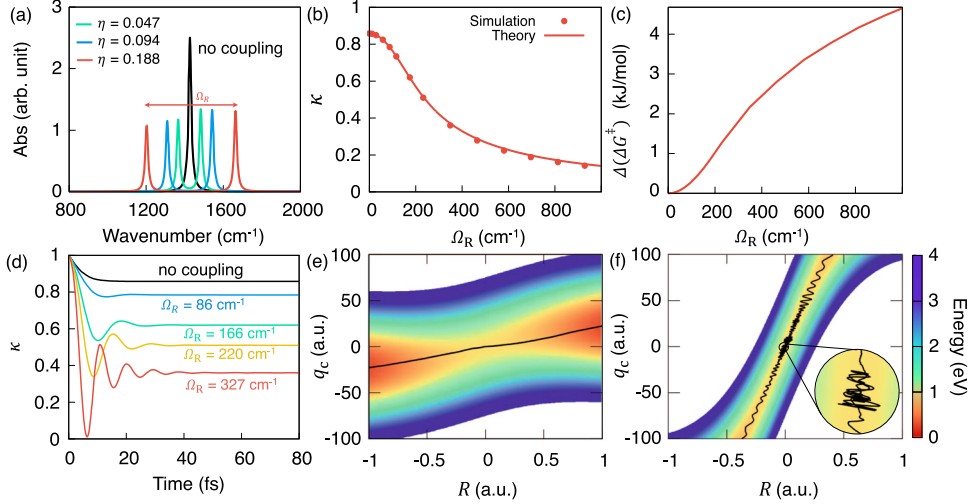

**Fig. 2 Decrease in the rate constant as increasing light-matter couplings. a** Infrared absorption spectrum by changing the normalized light-matter coupling strength $\eta$ (see Eq (2)). **b** The transmission coefficient $\kappa$ (under the limit $t \to t_p$) under various light-matter coupling strength (indicated by $\Omega_R$) at the resonant frequency $\hbar\omega_c = \hbar\omega_0 = 0.1706$ eV. **c** The "effective change" of the Gibbs free energy barrier $\Delta(\Delta G^{\ddagger})$ with respect to the coupling strength $\Omega_R$ at 300 K. **d** Time-dependent transmission coefficient $\kappa(t)$ at various light-matter coupling strengths at the resonant frequency $\hbar\omega_c = 0.1706$ eV. **e, f** Cavity Born–Oppenheimer surfaces $\hat{V}_{CBO}(R, q_c)$ at $\eta = 0.047$ and $\eta = 0.376$, respectively, with representative reactive trajectories indicated with black solid lines.

(see "Methods") as follows

$$k = \frac{\sqrt{-(\Omega_-^{\ddagger})^2}}{\omega_b} \cdot \frac{\omega_0}{2\pi} e^{-\beta E_b} \equiv \kappa_{GH} \cdot k_{TST}, \quad (5)$$

where $\Omega_-^{\ddagger}$ is the unstable imaginary normal-mode frequency on top of the barrier $((\Omega_-^{\ddagger})^2 < 0)$ and $\omega_b$ is the barrier frequency with $M\omega_b^2 = -\frac{\partial^2 E(R)}{\partial R^2}|_{R_{\ddagger}}$ as the curvature of the reaction barrier. The transmission coefficient $\kappa_{GH}$ for this simple 2D model is (see details in the "Methods" section as well as Supplementary Note 4)

$$\kappa_{GH} = \frac{1}{\omega_b} \left[ \frac{1}{2} \left( -\Delta\omega_{\ddagger}^2 + \sqrt{(\Delta\omega_{\ddagger}^2)^2 + 4\omega_b^2\omega_c^2} \right) \right]^{\frac{1}{2}}, \quad (6)$$

where $\Delta\omega_{\ddagger}^2 \equiv \omega_c^2 - \omega_b^2 + \frac{C_{\ddagger}^2}{\omega_c^2}$, with $C_{\ddagger} = \sqrt{\frac{2\omega_c}{M\hbar}}\chi \cdot \mu_{\ddagger}'$ characterizes the effective coupling between photonic coordinate $q_c$ and nuclear reaction coordinate $R$ in the transition state region, and $\mu_{\ddagger}' = \frac{\partial\mu}{\partial R}|_{R_{\ddagger}}$ is the slope of the dipole moment on the dividing surface $R_{\ddagger}$. Based on Eq. (6), one can derive that $\kappa_{GH}$ will have a minimum when

$$\omega_c = -\frac{\hbar}{2}\tilde{\eta}^2\mu_{\ddagger}'^2 + \frac{1}{2}\sqrt{\hbar^2\tilde{\eta}^4\mu_{\ddagger}'^4 + 4\omega_b^2} \quad (7)$$

where $\tilde{\eta} = \frac{\chi}{\sqrt{M\hbar\omega_c}}$ (note that the normalized coupling strength is $\eta = \mu_0'\sqrt{\frac{\hbar}{2\omega_0}}\tilde{\eta}$). Note that the term $\hbar\tilde{\eta}^2\mu_{\ddagger}'^2$ arises from $\frac{C_{\ddagger}^2}{\omega_c^2} = 2\hbar\omega_c \cdot \tilde{\eta}^2\mu_{\ddagger}'^2$ which is related to the light-matter coupling strength and appears within $\Delta\omega_{\ddagger}$ in Eq. (6). Further, the term $\frac{C_{\ddagger}^2}{\omega_c^2}$ also appears as an amplitude to the photonic friction kernel (see Supplementary Eq. 51) in the generalized Langevin equation. We emphasize that Eq. (7) provides a resonant effect of the reaction rate constant (through the transmission coefficient) when the cavity frequency $\omega_c$ is tuned to the above value. When $\eta$ is small (such that $\tilde{\eta}^4\mu_{\ddagger}'^4 \ll 4\omega_b^2$), the resonant frequency is close to the original barrier frequency $\omega_b$. As the coupling strength $\eta$ increases, the minimum will be shifted to the low-frequency region (with a redshift). Note that this resonant condition to achieve a minimum in $\kappa$ (Eq. (7)) is different from the one (which is $\omega_c = \omega_0$) to form the vibrational polariton in Eq. (2). When explicitly considering the

vibrational coupling to $R$ within $\hat{H}_{vib}$, $\kappa_{GH}$ has a more complicated expression as shown in Supplementary Note 4. Nevertheless, the presence of $\hat{H}_{vib}$ does not change the resonant condition in Eq. (7) (see Supplementary Fig. 1). The detailed procedure for obtaining the transmission coefficient as well as several key parameters of our current model system is provided in the "Methods" section.

**Central hypothesis.** With the above analysis, we conjecture that the cavity radiation mode inside the optical cavity is effectively acting as a "solvent" degree of freedom (DOF) that is coupled to the molecular reaction coordinate $R$, such that the presence of photonic coordinate enhance the recrossing of the reaction coordinate and reduces the transmission coefficients. A similar phenomenon is commonly referred to as the "dynamical caging" regime in simple organic reactions[30,47,48] and enzymatic catalysis[49–51], which have been successfully explained by the GH theory. Due to the low frequency of the photonic cavity mode (which is in the same range of the vibrational frequencies), we treat both $R$ and $q_c$ as the classical DOFs[17–19], and use the GH theory to explore the role of the cavity mode on reaction dynamics.

**Decreasing $\kappa$ as increasing $\Omega_R$.** Figure 2 presents the influence of increasing light-matter coupling $\eta$ (thereby increasing $\Omega_R$) on the reaction transmission coefficient $\kappa$ with the model Hamiltonian presented in Eq. (1). Figure 2a presents the IR spectrum computed based on the quantum light-matter interaction (Eq. (13) in "Methods"). The numerically exact Rabi splitting $\hbar\Omega_R$ is slightly deviated from $2\hbar\omega_c \cdot \eta$ (as indicated by Eq. (2)) due to the linear approximation ($\mu(R) \approx \mu_0 + \mu_0'R$) used in Eq. (2) (see the Supplementary Fig. 2). Figure 2b presents the transmission coefficient $\kappa$ obtained from direct numerical simulations (Eq. (4) under the $t \to t_p$ limit) as well as from the GH theory (solid lines) $\kappa_{GH}$ (by solving Eq. (12) in "Methods"). The GH theory quantitatively agrees with the results from the direct numerical simulations. With an increasing Rabi splitting $\Omega_R$, the transmission coefficient $\kappa$ decreased by almost one order of magnitude, whereas the TST rate $k_{TST}$ remains unchanged (due to the unchanged barrier height in the PF QED Hamiltonian). These numerical results corroborate our hypothesis that the suppression of chemical rate

originates from $\kappa$, which closely resembles the experimental result (e.g., Fig. 3D in ref. [22]).

Figure 2c presents another interesting result in this work. For the PF Hamiltonian description that explicitly includes the DSE term, there is no change in $k_{TST}$ because there is no change of potential energy barrier (see Fig. 1d) nor free energy barrier[18]. The only change in the rate comes from $\kappa$. However, one can back out the "effective change" of the free energy barrier height due to the changing $\kappa$. To this end, we use the Eyring rate equation (see "Methods") to convert the change of rate from $\kappa$ into an effective $\Delta(\Delta G^{\ddagger})$. The 4 times decrease in $\kappa$ presented in Fig. 2b results in ~4 kJ/mol change in "effective" $\Delta(\Delta G^{\ddagger})$ in Fig. 2c at ~700 cm$^{-1}$ of $\Omega_R$. We emphasize that this is not the real change of the free energy barrier height, but rather an "effective" change of $\Delta G^{\ddagger}$ according to the change of $\kappa$ based on our theoretical analysis. Interestingly, the experimentally measured results of $\Delta(\Delta G^{\ddagger})$ (Fig. 3C in ref. [22], for example) closely resemble our theoretical finding in Fig. 2c, with the key difference that our theoretical results suggest that these are not the actual free energy barrier changes, but entirely due to the change of $\kappa$, i.e., kinetics. Note that if one hypothesizes that an unknown mechanism to force the upper or lower vibrational polariton states to be a gateway of VSC polaritonic chemical reaction[52], then the activation energy change should shift linearly[18] with $\Omega_R$. The experimental results, on the other hand, demonstrate a non-linearity of reaction barrier[22]. Our theory indicates a non-linear increase of the "effective" $\Delta(\Delta G^{\ddagger})$ as increasing $\Omega_R$ due to the change of $\kappa$, closely resembles the experimental discoveries (Fig. 3C, D in ref. [22]).

Figure 2d presents the time-dependent simulation of the transmission coefficient $\kappa(t)$ defined in Eq. (4). With an increasing light-matter coupling hence a larger $\Omega_R$, the plateau value of $\kappa(t)$ keeps decreasing, and at the same time, $\kappa(t)$ becomes more oscillatory. This is a typical behavior of the reaction dynamics in the solvent caging regime[53]. As the coupling between $q_c$ and $R$ increase, the non-Markovian dynamics of $q_c$ can significantly influence the recrossing dynamics of the reaction coordinate $R$, from the "non-adiabatic" limit of a weak coupling regime to the "dynamic caging" of a strong coupling regime[39,53].

To clearly demonstrate the difference between these two regimes, we further present the Cavity BO surface $V_{CBO} = H - \frac{p^2}{2M} - \frac{p_c^2}{2} - H_{vib} = E(R) + \frac{1}{2}\omega_c^2(q_c + \sqrt{\frac{2}{\hbar\omega_c^3}}\chi \cdot \mu(R))^2$ along $R$ and $q_c$ in panel (e) and (f), with a representative reactive trajectory on top (black solid curve). Figure 2e presents a typical non-adiabatic case of the GH theory. When the instantaneous friction is weak ($\frac{|C|}{\omega_c} \ll \omega_b$), the GH theory becomes a model of non-equilibrium solvation, where the friction from the photonic coordinate $q_c$ does not severely impede the transitions[53]. In this case, the transmission coefficient remains close to the case without the cavity (black curve in Fig. 2d), and the reactive trajectory crosses the barrier without much influence from $q_c$. Figure 2f presents a typical "dynamical caging" regime of the GH theory, where the instantaneous friction from $q_c$ to $R$ is strong ($\frac{|C|}{\omega_c} \gg \omega_b$), such that the reaction coordinate $R$ becomes trapped in a narrow "solvent cage" on the barrier top[53]. At longer times, the bath relaxations of $\hat{H}_{vib}$ allow the $R$ to move away from the barrier top, but at shorter times, the reaction coordinate $R$ oscillates within the cavity-induced "solvent" cage[54]. The trajectory recrosses the dividing surface ($R_{\ddagger} = 0$) many times, resulting in oscillations of $\kappa(t)$ at a short time and with a small plateau value of $\kappa(t)$ at $t_p$ (see red curve in Fig. 2d). Similar dynamical caging effects from the solvent have been extensively studied in simple organic reactions (S$_N$1 and S$_N$2)[30,47,48] and enzymatic reactions[49–51], where the solvent dynamics significantly influences the reaction rate constant[39,40,53,55,56]. Here, the cavity photonic coordinate $q_c$ acts like a "solvent coordinate", and for strong couplings between $q_c$ and $R$, the system exhibits the dynamical caging effect which effectively slows down the reaction rate constant. This is our theoretical explanation for the observed suppression of the rate constant for VSC polariton chemical reactions[20,21,23,24].

**The origin of the resonant effect.** Figure 3a presents the transmission coefficient $\kappa$ (when $t \to t_p$) as a function of the photon

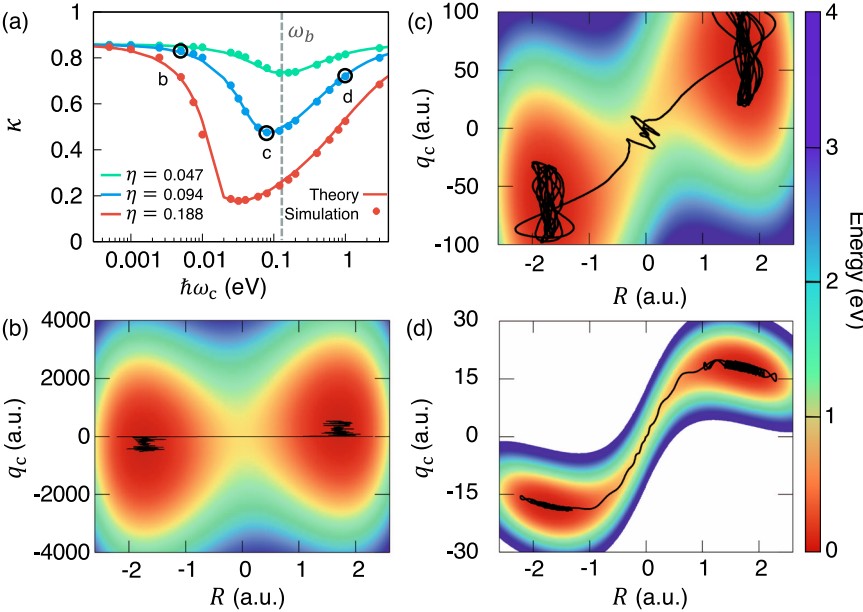

**Fig. 3 Resonant effect in vibrational strong coupling regime of polariton chemistry. a** Transmission coefficient $\kappa$ as a function of the photon frequency at three different values of the coupling strength $\eta$. **b–d** The cavity Born–Oppenheimer surfaces $V_{CBO}(q_c, R)$ under the normalized coupling strength $\eta = 0.094$ (corresponding to the blue solid line in panel (**a**)) at the photon frequency **b** $\hbar\omega_c = 2.5$ meV, **c** 80 meV, and **d** 1.0 eV, with the representative reactive trajectories indicated with black solid curves.

frequency $\omega_c$ with three normalized coupling constant $\eta$ (defined in Eq. (2)). The results are obtained from the GH theory (solid line) as well as the direct numerical simulation of Eq. (4) (filled circles). One can clearly see a resonant behavior of $\kappa$ when changing the photon frequency, agreeing with the analytical result (Eq. (7)) of a simpler model. These findings in Fig. 3a closely resemble recent experimental results of desilylation reaction (Fig. 3A in ref. [21], Fig. 3B in ref. [20]), aldehyde/ketone Prins cyclization (Fig. 3 in ref. [24]), and enzymatic reaction in pepsin (Fig. 3C in ref. [23]). Note that under a relatively small light-matter coupling $\eta = 0.047$ (green), the resonant frequency that gives a minimal $\kappa$ is close to $\omega_b$, which is also close to the reactant equilibrium frequency of the reactant $\omega_0$ in our SM model. For the parameter regime $\eta < 0.1$ (not entering into the USC), we find that the resonant condition (based on Eq. (7)) is close to $\omega_b$.

Note that experimentally, one often plot the cavity frequency-dependent reaction kinetics against the absorption curve of vibrational polariton. With our theoretical understanding and model calculations, we conclude that these two resonant behavior have two different origins and resonant frequencies. The resonant condition observed in the IR spectrum for Rabi Splitting requires $\omega_c = \omega_0$, whereas the resonant effects for a minimum of the rate constant require $\omega_c \approx \omega_b$. However, it is possible for a given molecular system which has $\omega_0 \approx \omega_b$. For example, in a theoretical work (at the level of MP2 perturbation theory) by Merkel and co-workers[57], a well-studied $S_N2$ reaction ($CH_3F + H^- \rightarrow CH_4 + F^-$) has a $\omega_b = 975.5$ cm$^{-1}$, which is close to one ground-state vibrational frequency $\omega_0 = 978.7$ cm$^{-1}$. In fact, this reaction could be also an ideal one subject to future investigations of VSC modifications of reactivities. On the other hand, there are also cases where $\omega_0$ and $\omega_b$ are different. For example, in an $S_N2$ reaction involving a Si–C bond cleavage in 1-phenyl-2-trimethyl-silylacetylene, we find (using the geometries reported in ref. [58] at the same level of electronic structure theory) that the computed imaginary barrier frequency to be $\omega_b \approx 74$ cm$^{-1}$, whereas the Si–C stretching frequency in the reactant well[58] is $\omega_0 \approx 860$ cm$^{-1}$.

When increasing the coupling strength to the USC regime ($0.1 < \eta < 1.0$), the resonant frequency is significantly red-shifted from $\omega_b$. For example, when $\eta = 0.188$ (red curve), the resonant condition for reaching a minimum value of $\kappa$ is 25 meV. Nevertheless, in the range of 10 meV $< \hbar\omega_c <$ 100 meV, $\kappa$ remains a very low value around 0.2, similar to the value at $\omega_c = \omega_b$. This red-shift of resonant frequency at which the rate constant is most significantly reduced has not been observed experimentally. Our theory predicts that if VSC experiments can reach the ultra-strong coupling regime, then the resonant frequency will be significantly shifted.

The origin of this resonant behavior in VSC chemical reaction rate constant (as indicated in Eq. (7)) can also be intuitively understood by examining representative trajectories (black solid curves) on the cavity BO potential energy surfaces presented in Fig. 3b–d, with the black solid lines, indicate representative trajectories. At a very low frequency $\hbar\omega_c = 2.5$ meV shown in Fig. 3b, the photon coordinate essentially remains frozen compared to the dynamics of the reaction coordinate $R$ during the course of the reaction. As a result, under this frozen solvent limit, the transmission coefficient remains close to the no-coupling scenario. At $\hbar\omega_c = 80$ meV in Fig. 3c, with $\frac{|C|}{\omega_c} \gg \omega_b$, the light-matter interactions lead to the dynamical caging of the reaction coordinate at the barrier top, leading to significant decrease in the transmission coefficient $\kappa_{GH}$. When the photon frequency is further increased ($\hbar\omega_c = 1$ eV), the reactant and the product wells become separated with a narrow channel as shown in Fig. 3d. At such a high photon frequency, the reactive channel connecting the reactant and product becomes extremely narrow[59]

(much narrower than the usual dynamical caging scenario depicted in Fig. 3c or Fig. 2f), such that the reactive trajectories almost follow a straight path and is no longer caged near the dividing surface. As opposed to the dynamical caging regime, the transmission coefficient in Fig. 3d is less suppressed than the minimum $\kappa$ when the photon frequency is near $\omega_b$. Similar behavior of the reaction dynamics is also observed for the USC regime ($\eta = 0.188$ in Fig. 3), where the results are provided in Supplementary Fig. 6. Therefore, the suppression of the chemical kinetics through the dynamical caging effect by the photon mode is highly sensitive to the photon frequency, proving a plausible mechanism for explaining the resonant behavior[21,23,24] of the reaction rate constant in VSC polariton chemistry.

## Discussion

In this work, we provide a theoretical explanation of the resonant VSC polariton chemistry reactivities. We demonstrate that the resonant suppression of the reaction rate constant using the analytical GH rate theory as well as performing numerical calculations for a SM model molecular system coupled to a single-radiation mode inside an optical cavity. As opposed to the previous theoretical studies[17–19,27] that only focuses on the transition state theory, our investigation suggests that the coupling between a cavity photonic mode and a molecule leads to the suppression of the transmission coefficient of the rate constant, exhibiting the resonant behavior which can be explained by simple GH rate theory. Through both analytical theory and numerical simulations, we demonstrate that the cavity photon mode acts like a "solvent" DOF which influences the chemical kinetics and leads to the suppression of the transmission coefficient. Such an effect is purely dynamical and is not captured within a simple transition state theory.

Further, our theoretical hypothesis provides a plausible explanation to the observed resonant effects of the electronically adiabatic ground-state reactions coupled to an optical cavity, whereas previous theoretical studies[17–19,27] based upon a simple TST always conclude a frequency-independent VSC rate constant. The suppression of the rate constant is sensitive to the photon frequency, such that the maximum suppression is achieved when the photon frequency is close to the barrier frequency in the vibrationally strong coupling regime when $\eta < 0.1$ and is red-shifted in the vibrationally ultra-strong coupling regime when $0.1 < \eta < 1$. Our results indicate that the resonant condition for achieving the Rabi splitting in the IR spectrum and the resonant condition for achieving a maximum suppression of the reaction rate constant are fundamentally different. While the former is related to the frequency of the reactant, the latter is related to the top of the barrier frequency and the molecule-cavity coupling strength.

We want to remind the reader that the present work is limited to a single molecule coupled to a single radiation mode, whereas the experimentally observed frequency-dependent modification of the chemical kinetics in the collective coupling regime. It was suggested in ref. [19] that the resonant effects will disappear under the $N \rightarrow \infty$ limit, where $N$ is the number of molecules coupled to the cavity. However, we believe that the effect we have seen will be present under the few $N$ limit (in the current paper, $N = 1$). Whether our current theory can also be extended to the collective regime remains an open question. The present formalism can be extended to include cavity losses and their impact on the caging effect. The VSC experiments often use low-quality factor cavities, where the cavity loss should also be explicitly included. Interestingly, these far-field modes which are responsible for the cavity loss can be modeled as dissipative modes coupled to the quantized modes of a cavity, providing an additional dissipative environment for the hybrid system. The frequency dependence as

well as the collective phenomenon might also emerge when cavity loss is explicitly included[60].

Overall, our work emphasizes the importance of the dynamical effect induced by the cavity photon modes on chemical kinetics to explain new chemical reactivities observed in recent experimental studies on vibrational strong coupling of molecules and cavity. Future investigations will focus on understanding the collective VSC reactivities by coupling many molecules with the cavity[18,19].

## Methods

**Pauli–Fierz QED Hamiltonian.** The minimal coupling QED Hamiltonian in the Coulomb gauge (the "p · A" form) is expressed as

$$\hat{H}_C = \sum_j \frac{1}{2m_j}(\hat{\mathbf{p}}_j - z_j\hat{\mathbf{A}})^2 + \hat{V}(\hat{\mathbf{x}}) + \hat{H}_{ph}, \qquad (8)$$

where the sum is performed over all charged particles, including electrons and nuclei, $m_j$ and $z_j$ are mass and charge for particle $j$, respectively, and $\hat{\mathbf{p}}_j = -i\hbar\nabla_j$ is the canonical momentum operator. Further, under the Coulomb gauge, $\nabla \cdot \mathbf{A} = 0$, the vector potential becomes purely transverse $\hat{\mathbf{A}} = \hat{\mathbf{A}}_\perp$. Under the long-wavelength approximation, $\hat{\mathbf{A}} = \mathbf{A}_0(\hat{a} + \hat{a}^\dagger) = \mathbf{A}_0\sqrt{2\omega_c/\hbar}\,\hat{q}_c$, where $\mathbf{A}_0 = \sqrt{\hbar/2\omega_c\varepsilon_0\mathcal{V}}\cdot\mathbf{e}$, with $\mathcal{V}$ as the quantization volume inside the cavity, $\varepsilon_0$ as the permittivity, and $\mathbf{e}$ is the unit vector of the field polarization. Using the Power–Zienau–Woolley (PZW) gauge transformation operator[61,62] $\hat{U} = \exp[-\frac{i}{\hbar}\hat{\boldsymbol{\mu}}\cdot\hat{\mathbf{A}}] = \exp[-\frac{i}{\hbar}\hat{\boldsymbol{\mu}}\cdot\mathbf{A}_0(\hat{a}+\hat{a}^\dagger)]$, as well as a unitary transformation operator $\hat{U}_\phi = \exp[-i\frac{\pi}{2}\hat{a}^\dagger\hat{a}]$, the Pauli–Fierz (PF) Hamiltonian is obtained as

$$\hat{H}_{PF} = \hat{U}_\phi\hat{U}\hat{H}_C\hat{U}^\dagger\hat{U}_\phi^\dagger = \hat{H}_M + \frac{1}{2}\hat{p}_c^2 + \frac{1}{2}\omega_c^2\left(\hat{q}_c + \sqrt{\frac{2}{\hbar\omega_c}}\hat{\boldsymbol{\mu}}\cdot\mathbf{A}_0\right)^2, \qquad (9)$$

where the matter Hamiltonian is $\hat{H}_M = \hat{T}_R + \hat{H}_{el} \equiv \hat{T}_R + \hat{T}_r + \hat{V}$, with $\hat{T}_R$ and $\hat{T}_r$ representing the nuclear and electronic kinetic energy, respectively, and $\hat{V}$ representing the Coulomb interaction potential among all charged particles (electrons and nuclei), and $\hat{H}_{el}$ is the electronic Hamiltonian. The detailed derivation is provided in Supplementary Note 1. The presence of DSE (the $A_0^2$ term in Eq. (9)) is necessary in order to have a Gauge invariant Hamiltonian[63,64] and it has shown to be crucial for an accurate description of light–matter interactions under the dipole gauge[63–65]. Projecting $\hat{H}_M$ and $\hat{\mu}$ in the ground electronic state $|\Psi_g\rangle$ (which is obtained by solving $\hat{H}_{el}|\Psi_g\rangle = E(R)|\Psi_g\rangle$), we obtain the model Hamiltonian in Eq. (1).

**Grote–Hynes rate theory.** In multidimensional transition state theory, the reactant to product rate constant is given as[40,44–46]

$$k = \frac{1}{2\pi}\frac{\prod_{i=1}^N \Omega_i^0}{\prod_{i=2}^N \Omega_i^\ddagger}e^{-\beta E_b}, \qquad (10)$$

where $\{\Omega_i^0\}$ are normal-mode frequencies of the Hamiltonian in the reactant well, and $\{\Omega_2^\ddagger, ..., \Omega_N^\ddagger\}$ are the stable normal-mode frequencies at the barrier, such that $\Omega_i^{\ddagger 2} > 0$ for $i > 1$, and $\Omega_1^{\ddagger 2} < 0$ is the imaginary frequency of the transition state.

Considering a simplified (classical) model of the molecule-cavity hybrid system, $H - H_{vib} = \frac{P^2}{2M} + E(R) + \frac{p_c^2}{2} + \frac{1}{2}\omega_c^2(q_c + \sqrt{\frac{2}{\hbar\omega_c^3}}\chi\cdot\mu(R))^2$ which only contains two DOFs $\{q_c, R\}$ (and are viewed as classical DOFs), the normal-mode frequencies at $R_0$ are $\Omega_\pm^2 = \frac{1}{2}(\omega_0^2 + \frac{C_0^2}{\omega_c^2} + \omega_c^2) \pm \frac{1}{2}\sqrt{(\omega_0^2 + \frac{C_0^2}{\omega_c^2} + \omega_c^2)^2 - 4\omega_c^2\omega_0^2}$, where $C_0 = \sqrt{\frac{2\omega_c}{M\hbar}}\chi\cdot\mu_0'$. The normal-mode frequencies at $R_\ddagger$ are

$\Omega_\pm^{\ddagger 2} = \frac{1}{2}(-\omega_b^2 + \frac{C_\ddagger^2}{\omega_c^2} + \omega_c^2) \pm \frac{1}{2}\sqrt{(-\omega_b^2 + \frac{C_\ddagger^2}{\omega_c^2} + \omega_c^2)^2 + 4\omega_c^2\omega_b^2}$, where

$C_\ddagger = \sqrt{\frac{2\omega_c}{M\hbar}}\chi\cdot\mu_\ddagger'$. Details of the derivation of these normal-mode frequencies are provided in Supplementary Note 2. Using these normal-mode frequencies, the rate constant in Eq. (10) for the $\hat{H} - \hat{H}_{vib}$ model is expressed as $k = \frac{1}{2\pi}\frac{\Omega_+\Omega_-}{\Omega_+^\ddagger}e^{-\beta E_b}$, where $E_b$ is the energy barrier. Using the fact that $(\Omega_+^\ddagger\Omega_-^\ddagger)^2 = -\omega_b^2\omega_c^2$ and $(\Omega_+\Omega_-)^2 = \omega_0^2\omega_c^2$ (see the general proof in ref. [42]), the rate constant can be further expressed as follows

$$k = \frac{1}{2\pi}\sqrt{-(\Omega_-^\ddagger)^2}\frac{\omega_0\omega_c}{\omega_b\omega_c}e^{-\beta E_b} = \frac{\lambda}{\omega_b}\cdot\frac{\omega_0}{2\pi}e^{-\beta E_b} \equiv \kappa_{GH}\cdot k_{TST}, \qquad (11)$$

where $k_{TST} = \frac{\omega_0}{2\pi}e^{-\beta E_b}$, $\kappa_{GH} = \frac{\lambda}{\omega_b}$ is the transmission coefficient in the GH theory, $\lambda = \sqrt{-(\Omega_-^\ddagger)^2}$, which is Eq. (6). Same procedure can also be used to derive the expressions[46] of the $\kappa_{GH}$ for the model Hamiltonian in Eq. (1). Alternatively, one can

derive the transmission coefficient $\kappa_{GH}$ from the equation of motion[39,56], with the details provided in Supplementary Note 4.

When considering the phonon bath $\hat{H}_{vib}$ under the Markovian limit (while consider $q_c$ as the non-Markovian coordinate), $\lambda$ can be obtained by solving the following equation

$$\lambda^4 + \frac{\zeta}{M}\lambda^3 + \left(\omega_c^2 - \omega_b^2 + \frac{C_\ddagger^2}{\omega_c^2}\right)\lambda^2 + \frac{\zeta}{M}\omega_c^2\lambda - \omega_c^2\omega_b^2 = 0, \qquad (12)$$

where $\kappa_{GH} = \frac{\lambda}{\omega_b}$. We consider a bath friction coefficient $\zeta = 400\,\text{cm}^{-1}$ according to the spectral density $J(\omega)$. The detailed derivations of Eq. (12), as well as the assessment of the validity of the Markovian limit of the phonon bath are provided in Supplementary Note 4.

**Model molecular Hamiltonian.** The potential energy surface (PES) and permanent dipole moment are taken from a SM model[66], which is illustrated in Fig. 1. The SM model is a one-dimensional molecular system that describes a proton-coupled electron transfer reaction between a donor and an acceptor ion. The model consists of a transferring proton, an electron, and two fixed ions. The molecular Hamiltonian is $\hat{H}_M = \frac{\hat{P}^2}{2M} + \hat{H}_{el} + \hat{H}_{vib}$, where $M$ is the mass of the nuclei (proton in the SM model), $\hat{H}_{el} = \hat{T}_r + \hat{V}_{eN} + \hat{V}_{NN}$ is the electronic Hamiltonian, where $\hat{T}_r = \hat{p}_r^2/2m_e$ represents the kinetic energy operator of the electron with mass $m_e$, $\hat{V}_{eN}$ describes the interaction between the electron and the three nuclei, and $\hat{V}_{NN}$ that describes the Coulomb repulsion between the proton and the static ions. The resulting PES $E(R) = \langle\Psi_g(R)|(\hat{H}_M - \hat{T}_R)|\Psi_g(R)\rangle$ and the permanent dipole moment $\mu(R) = \langle\Psi_g(R)|\hat{\mu}|\Psi_g(R)\rangle$ are shown in Figs. 1b and 1c, respectively. The details of this model, as well as the numerical procedure to obtain the ground-state potential and dipole are provided in Supplementary Note 3 and Supplementary Note 5.a.

In addition, $\hat{H}_{vib} = \sum_k \frac{P_k^2}{2M_k} + \frac{1}{2}M_k\omega_k^2(R_k + \frac{c_k}{M_k\omega_k^2}\cdot R)^2$ is the vibrational system-bath Hamiltonian that describes the interactions between reaction coordinate $R$ and other vibrational phonon modes in the molecule. The coupling constant $c_k$ and the frequency $\omega_k$ is characterized by an ohmic spectral density $J(\omega) = \frac{\pi}{2}\sum_k \frac{c_k^2}{M_k\omega_k}\delta(\omega - \omega_k) = \zeta\omega e^{-\omega/\omega_p}$, with a characteristic phonon frequency $\omega_p$ and a friction constant $\zeta$. In Table 1, we outline several key parameters in our model system, whereas the full details are provided in Supplementary Note 3.

**Numerical simulation of $\kappa(t)$.** All simulations were performed under $T = 300\,\text{K}$ by evolving the classical dynamics governed by $H(R, q_c)$ in Eq. (1). Langevin dynamics is used to model the influence of $H_{vib}$ on the light-matter hybrid system, whereas $q_c$ is explicitly propagated in time and treated as a non-Markovian "solvent" DOF. The friction constant in the Langevin dynamics was chosen to be $\zeta = 400\,\text{cm}^{-1}$ according to the spectral density of the $\hat{H}_{vib}$ (see details in Supplementary Note 3 and 4). The time step used in the simulation is $dt = 4\,\text{a.u.}$, which was carefully checked to produce stable integration for all simulations. From a long constraint MD trajectory on the dividing surface $R_\ddagger = 0$, the constrained configurations $\{q_c, R_\ddagger\}$ are sampled for every 270 fs along that constrained trajectory. A total of 100,000 trajectories are released from the dividing surface, with the initial velocities randomly sampled from the classical Maxwell–Boltzmann distribution. Each of the sampled configuration is propagated for 200 fs, which guaranteed that the flux-side correlation function would plateau. The flux-side correlation function in Eq. (4) is computed through the ensemble average. Details of the numerical simulation procedure are provided in Supplementary Note 5.d.

**Effective $\Delta(\Delta G^\ddagger)$.** To account for the "effective change" of the Gibbs free energy barrier $\Delta(\Delta G^\ddagger)$ corresponding to the changes in $\kappa$, we consider the Eyring rate equation $k = \frac{k_B T}{h}e^{-\frac{\Delta G^\ddagger}{k_B T}}$, and thus $\Delta G^\ddagger = -\frac{1}{\beta}\ln(2\pi\beta\cdot k)$. With $k = \kappa\cdot k_{TST}$, we can rewrite the above $\Delta G^\ddagger$ as $\Delta G^\ddagger = -\frac{1}{\beta}\ln\kappa - \frac{1}{\beta}\ln 2\pi\beta k_{TST}$. Because $k_{TST}$ is a constant at any coupling strength and cavity frequency and is the same for bare molecular case, the effective $\Delta(\Delta G^\ddagger)$ solely depends on the change of $\kappa$. The change of free energy barrier compared to the bare molecular reaction (with $\kappa_0$ and $\Delta G_0^\ddagger$) is then $\Delta(\Delta G^\ddagger) = \Delta G^\ddagger - \Delta G_0^\ddagger = -\frac{1}{\beta}\ln\frac{\kappa}{\kappa_0}$, which is used to compute the value presented in Fig. 2c.

### Table 1 Key parameters of model.

| $\hbar\omega_0$ (meV) | $\hbar\omega_b$ (meV) | $\mu_0'$ (a.u.) | $\mu'$ (a.u.) |
|---|---|---|---|
| 170.6 | 162.05 | 0.225 | −1.887 |

The frequency of the reactant well $\omega_0$ and the top of the barrier $\omega_b$, as well as the derivative of dipole moment at the equilibrium geometry $\mu_0'$ and on the dividing surface $\mu_\ddagger'$.

**Absorption spectrum**. We employ a simple approach[67] to compute the absorption spectrum of the molecule-cavity hybrid system. The absorption cross section $\sigma(\mathcal{E})$ as a function of excitation energy $\mathcal{E}$ is expressed[67,68] as follows

$$\sigma(\mathcal{E}) = \frac{4\pi\mathcal{E}}{c} \text{Im} \left[ \sum_{\nu \neq 0} \frac{|\langle \Phi_\nu | \mu(R) | \Phi_0 \rangle|^2}{\mathcal{E}_\nu - \mathcal{E}_0 - \mathcal{E} - i\varepsilon} \right], \quad (13)$$

where $\varepsilon$ a phenomenological width parameter that accounts for the broadening of the absorption spectrum, and $c$ is the speed of the light. Further, $\mathcal{E}_\nu$ is the energy of the $\nu_{\text{th}}$ vibrational polaritonic state of $\hat{H}_{\text{vpl}} = \frac{\hat{P}^2}{2M} + E(\hat{R}) + \frac{1}{2}\hat{p}_c^2 + \frac{1}{2}\omega_c^2(\hat{q}_c + \frac{A_0\mu(\hat{R})}{\sqrt{\hbar\omega_c}})^2$, and $\mathcal{E}_0$ is ground-state vibrational polaritonic eigenenergy. Details of this calculation are provided in Supplementary Note 5.b.

## Data availability
The data that support the plots within this paper and other findings of this study are available from the corresponding authors upon a reasonable request.

## Code availability
The source code for simulating the polariton eigenstates, the absorption spectrum, and the transmission coefficients are available from the corresponding authors upon a reasonable request.

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

## Acknowledgements

This work was supported by the National Science Foundation CAREER Award under Grant No. CHE-1845747 and "Enabling Quantum Leap in Chemistry" program under a Grant number CHE-1836546, as well as by a Cottrell Scholar award (a program by of Research Corporation for Science Advancement). Computing resources were provided by the Center for Integrated Research Computing (CIRC) at the University of Rochester.

## Author contributions

X.L., A.M. and P.H. designed the research. X.L. performed the transmission coefficient simulations. X.L. and A.M. set up the model for molecule-cavity system and performed the absorption spectrum simulations. A.M. performed analytical derivations of the GH rate theory. X.L. and A.M. contributed equally on analyzing the data. X.L., A.M. and P.H. wrote the manuscript.

## Competing interests

The authors declare no competing interests.
