## [Peer Review File · Nature Communications]

REVIEWER COMMENTS

Reviewer #1 (Remarks to the Author):

The paper contains a study on vibrational strong light-matter coupling in the emerging field of polaritonic chemistry using the Shin-Metiu model coupled to optical cavity. Several recent experiments have demonstrated recently that strong vibrational coupling can alter chemical reactions (e.g. Refs 20-26 in the paper), however a good theoretical explanation for this effect still remains elusive. Even worse, there are several recent publications that argue that no sizable and no resonant effect on chemical reactivity in these systems should be seen or expected (e.g. Refs. 17-19). The paper of Li et al. sheds some light on these experiments. The authors show for the first time under which circumstances resonant and sizable effects of vibrational strong coupling on chemical reactions can appear. In my opinion these novel findings warrant publication in Nature Communications.

I have few points and comments, after those have been sufficiently addressed I can recommend publication in Nature Communications.

1. The setup of the Shin-Metiu model is chosen such that the barrier frequency and the vibrational frequency are close in energy. Do the findings in fig. 3 also hold if these two frequencies become very different? Are the two effects Rabi splitting and resonance with barrier frequency independent of each other?
2. The authors include a very brief statement that collective effects are currently under investigation. I want to ask nevertheless. For the Dicke model, it can be shown that the many electron system can be replaced by a single electron system with stronger interaction (famous \sqrt{N} scaling). Does a similar mapping also exist for the more complicated case that includes now a transition state? In this case similar behavior of the resonance condition with barrier frequency and the caging effect could be expected.
3. What is the physical interpretation of the different terms contributing to Eq. 6? I follow the reasoning that if ν is small, then cavity frequency \sim barrier frequency gives the resonance condition. However this may not be the case in general. Can one attribute a physical meaning to the first and second terms contributing to Eq.6? What is the physical significance of the detuning? Is it related to the dipole self-energy term?
4. The authors study a simple model including one single nuclear and one single electronic degree. Molecules are (unfortunately) much more complicated. How does the picture change if much more nuclear, and electronic degrees of freedom that can be correlated are present in the system?

5. Which electronic structure methods could be used for a first principle description of these effects? Are ab-initio methods for strong light-matter systems, such as in Phys. Rev. X 10, 041043 2020, Chem. Phys. 2018,509, 55-65, or Phys. Rev. Lett. 121, 113002 (2018) sufficient, or are there missing parts that need to be developed?

Reviewer #2 (Remarks to the Author):

The manuscript by Li, Mandal and Huo discusses a possible theoretical explanation for the observed phenomenon, that coupling to an optical cavity suppresses chemical reaction rates. The approach introduces a new idea, focusing on the dynamics of the reaction, and how the cavity degree of freedom can change the transmission coefficient. This is illustrated with a combination of analytical and numerical calculations, studying in detail a simple model of reaction kinetics.

I believe this is a significant paper. Prior to this work, there was no viable theoretical explanation of several features of the experimentally observed behavior. In particular, it was hard to explain why the reaction rate would show a resonant enhancement at a particular cavity frequency. The current manuscript provides a possible explanation of this observation. There are still some questions open. In particular, as noted by the authors, the question of how this idea generalizes for multiple molecules coupled to the same cavity. These are important questions, but I believe it is appropriate to publish the current manuscript without these. The manuscript presents and describes clearly a new idea that may be significant in understanding the effect of cavities on chemical reactions.

There are some presentational issues I would suggest the authors consider. These mostly arise from the fact this field is one involving both physicists and chemists. The manuscript as it stands is perhaps more targeted to an audience of chemists. Given the potential broad audience of this paper, it would be helpful to make sure it was accessible also to physicists.

Specific issues are:

* Main text page 3. There are references in this section to results "presented in the main text". This seems an odd phrase to use in the main text; perhaps this could be replaced by a more specific reference, e.g. "the $\hbar\Omega_R$ presented in Fig. 2"

* Main text Eq. 4, and supplemental Eq. 104. There are a few issues here:

Firstly, I am unclear what is meant by the time derivative of $R\ddagger$ in the denominator. $R\ddagger$ is defined as being the location of the dividing surface, so does not change with time. Should this be the time derivative of R instead?

Secondly, this formula is not necessarily familiar to physicists. It would be helpful to explain its meaning more clearly, in addition to referencing relevant papers.

Thirdly (related to the point above), I am somewhat confused why it is necessary to specify that the ensemble average is on the dividing surface when the flux contains a delta function that limits one to that dividing surface.

* Main text page 4, around Eq. 5. While I agree that this equation is the minimal result of this calculation, I think the existence of this simple form is important enough that it warrants a bit more explanation in the results, rather than having all details in the methods. I might suggest in particular Eq. 8 from the methods, and some of the following discussion. This would help make clear where this formula comes from --- i.e. that it comes from the form of a vibrational frequency Ω_{-}

* Main text, Page 5, there is a reference to Fig. 2e presenting a typical "dynamical caging" regime; I think this means Fig. 2f.

* Supplement, Figure 1 caption. This caption appears to use the phrase "Langevin limit" (three times) to refer to the Markovian limit.

* Supplement, inline equation between Eq. 92 and 93. The force on the nuclear coordinate seems to be missing a term from the derivative of $E(R)$.

Reviewer #3 (Remarks to the Author):

In "Resonant Theory of Vibrational Strong Couplings in Polariton Chemistry," Li and co-workers address the contemporary problem of vibrational-strong-coupling induced modification of adiabatic chemical kinetics. They expand upon previous cavity Born-Oppenheimer based approaches by considering VSC as having dynamical implications instead of only equilibrium ones. Within the frame of Grote-Hynes transition-state theory, the authors find that the rate constant does have a frequency dependence, which has not been explored in previous studies. They confirm their findings by performing a numerical simulation within the Shin-Metiu model. These analyses allow them to conclude that VSC produces an effect akin to the dynamical solvent caging, which modifies reaction rates in solution compared to in gas phase. Moreover, the authors explain the failure of previous approaches in terms of the resonance effects having a different nature from those that describe the absorption at polaritonic frequencies.

This is a remarkable finding that has evaded several attempts to explain Ebbesen and collaborators' results and is a sound step forward in understanding the observed effects. However, although this work shines a light on one of the main mysteries surrounding thermally activated reactions in infrared cavities, namely the frequency-dependent effects, it still falls short at addressing the collective nature of the phenomena, which has not only been confirmed experimentally but also determined to be an essential feature of the observations. The authors claim that their "theoretical hypothesis provides a plausible explanation to the observed resonant effects of the electronically adiabatic ground-state reactions coupled to an optical cavity" without discussing its obvious shortcomings: the transition dipole moment they assigned to the unstable mode is unrealistically high, which is an indication of the need of collective effects. Nonetheless, the theoretical basis upon which their results hold is applicable only when a single molecule (or possibly a few) couples to the cavity; in the collective regime, the arguments presented in refs 17, 19 and Chem. Phys. 535, 110767 (2020) (which the submission does not cite) would also nullify the findings of the manuscript at hand.

Discourse-wise, I am also uncomfortable with the authors' language referring to their finding as a resonant effect instead of a frequency-dependent one (in fact, a relatively low frequency one when compared to the IR cavity modes used in the Ebbesen experiments). Although the authors emphasize that the tuning role is different, I still think it might be misleading given the general understanding of this term in the community. Moreover, they found maximal effect without any resonance condition being strictly met. Thus, I suggest using the language of "frequency-dependent" highlights the true nature of the improvement of their approach over the previous ones. Finally, the authors give a hand-wavy explanation for the coincidental observation of resonant effects by claiming "however, it is possible for a given molecular system which has[sic] $\omega_0 \approx \omega_b$ ". For the sake of completeness, a reader might find useful a table that compares typical values of both frequencies for common SN2 reactants.

As explained above, I find this work relevant and timely, and I will gladly support its publication in Nature Communications if the authors adequately address my concerns.

Responses to the referees

We would like to thank the referees for their detailed, critical, and thorough reviews of our manuscript. We are glad to learn that they found our work “sheds some light on these (VSC) experiments, show for the first time under which circumstances resonant and sizable effects of vibrational strong coupling on chemical reactions can appear” and “warrant publication in Nature Communications.” (Reviewer 1), “presents and describes clearly a new idea that may be significant in understanding the effect of cavities on chemical reactions,” (Reviewer 2), and presents “remarkable finding that has evaded several attempts to explain Ebbesen and collaborators' results and is a sound step forward in understanding the observed effects” (Reviewer 3), and find it to be “relevant and timely, and will gladly support its publication in Nature Communications”.

We believe that the revised version of the manuscript has been significantly improved by addressing Reviewers' comments. Below, we discuss the comments by the Reviewers (in blue) and include the modifications to the manuscript (in red).

Reviewer #1:

Comment (Remarks to the Author): The paper contains a study on vibrational strong light-matter coupling in the emerging field of polaritonic chemistry using the Shin-Metiu model coupled to optical cavity. Several recent experiments have demonstrated recently that strong vibrational coupling can alter chemical reactions (e.g. Refs 20-26 in the paper), however a good theoretical explanation for this effect still remains elusive. Even worse, there are several recent publications that argue that no sizable and no resonant effect on chemical reactivity in these systems should be seen or expected (e.g. Refs. 17-19). The paper of Li et al. sheds some light on these experiments. The authors show for the first time under which circumstances resonant and sizable effects of vibrational strong coupling on chemical reactions can appear. In my opinion these novel findings warrant publication in Nature Communications.

I have few points and comments, after those have been sufficiently addressed I can recommend publication in Nature Communications.

Response : We thank the reviewer for the encouraging and insightful comments.

Comment 1. The setup of the Shin-Metiu model is chosen such that the barrier frequency and the vibrational frequency are close in energy. Do the findings in fig. 3 also hold if these two frequencies become very different? Are the two effects Rabi splitting and resonance with barrier frequency independent of each other?

Response: In this study, we find that the resonant condition for minimizing the transmission coefficient is when the photon frequency matches (or close to) the top of the barrier frequency, not when the photon frequency matches the vibrational frequency. Therefore, in our current theory, when the vibrational frequency is very different from the barrier frequency, the results of Fig.3 remain the same.

With our current model (which contains a single molecule- single cavity mode) the effects of Rabi splitting and resonance with barrier frequency are indeed independent. However, whether or not this independence exists in the case for many molecules coupled to the cavity remains an open question and is a subject of our ongoing research.

Since in the original text we had clarified the independence of Rabi splitting and the origin of the resonant effect multiple times, we did not make any new changes. **No change has been made.**

Comment 2. The authors include a very brief statement that collective effects are currently under investigation. I want to ask nevertheless. For the Dicke model, it can be shown that the many electron system can be replaced by a single electron system with stronger interaction (famous \sqrt{N} scaling). Does a similar mapping also exist for the more complicated case that includes now a transition state? In this case similar behavior

of the resonance condition with barrier frequency and the caging effect could be expected.

Response: As the reviewer correctly mentioned, the rabi-splitting scales with \sqrt{N} with N being the number of molecules coupled to the cavity. At this point, we do not see a similar simple argument that will lead to a \sqrt{N} scaling in the transmission coefficient with the present setup. The reason for that is because not all of the N molecules will reach the transition state *simultaneously* which leads to a collective coupling (of N molecules in the transition state configuration) to the cavity. This is the fundamental difference compared to the reactant ensemble of molecules, where N molecules are all in (or fluctuating around) the equilibrium reactant geometry, hence leading to collective coupling to the cavity. This is also why all of the current theoretical studies [eg, Li, Nitzan, Subtonik, J. Chem. Phys. 152, 234107 (2020)] based upon TST failed to explain the collective effects.

To address this comment, we have commented on the collective effect in our conclusion.

Changes: We have some comments in page 5 within a new paragraph that was added (also as a response to Reviewer 3) which starts with: “We want to remind the reader that the present work is limited to a single molecule coupled to a single radiation mode...”

Comment 3. What is the physical interpretation of the different terms contributing to Eq. 6? I follow the reasoning that if ν is small, then cavity frequency \sim barrier frequency gives the resonance condition. However this may not be the case in general. Can one attribute a physical meaning to the first and second terms contributing to Eq.6? What is the physical significance of the detuning? Is it related to the dipole self-energy term?

Response: The $(\tilde{\eta}\mu_{\pm}')^2$ part originates from $C_{\pm}^2/\omega_c^2 \propto (\tilde{\eta}\mu_{\pm}')^2$ which is related to the light-matter coupling strength and also appears as an amplitude to the photonic friction kernel (see Supplementary Eq. 51). We have added a comment below the said Equation (Eq. 7 in the modified manuscript). Also note that the equation had a typo, that it was missing a \hbar .

Change:

- We added a brief discussion on the origin of the term in the paragraph below Eq. 6 (now Eq. 7) which starts with: “Note, that the term $\hbar\tilde{\eta}\mu_{\pm}'^2$ arises from...”
- The Eqn.7 in the present manuscript has been modified to fix a missing \hbar :
“ $\omega_{\text{c}} = -\frac{\hbar}{2}\tilde{\eta}\mu_{\pm}'^2 + \frac{1}{2}\sqrt{\hbar^2\tilde{\eta}^4\mu_{\pm}'^4 + 4\omega_{\text{b}}^2}$ ”

Comment 4. The authors study a simple model including one single nuclear and one single electronic degree. Molecules are (unfortunately) much more complicated. How does the picture change if much more nuclear, and electronic degrees of freedom that can be correlated are present in the system?

Response: In the present work, we have included one nuclear degree of freedom (a reaction coordinate) and one cavity mode. Note that this nuclear degree of freedom is also coupled to a dissipative environment (see the system-bath Hamiltonian in Supplementary Eq. 29 in Section 4 of the Supplementary Information), which models the effects of the solvents as well as the effect of other modes within the molecule. However, these bath modes do not change the permanent dipole of the molecule, which is an assumption in our model calculation.

For more complex systems that involve multiple nuclear DOF (in the dividing surface as well as changing the dipole) and a single electronic state (ground state, for example) one can always compute the multidimensional TST rate (Eq. 9 in the Method Section) by writing down and then diagonalizing the Hessian matrix. From the perspective of numerical simulations, as long as there is a well-defined dividing surface, hence a clear definition of the reactant and product, one will always be able to carry out the numerical simulations to compute the transmission coefficient and TST rate (by computing the free energy profile). Ideally, by properly choosing a collective variable, computation of the reaction rate can effectively be reduced to a one-dimensional problem. At this point, we don't know what is the outcome of coupling many nuclear degrees of freedom within a molecule that may actively take part in a reaction. The present formalism can certainly be extended to investigate such scenarios. In this work, on the other hand, our aim has been to capture the *essential physics* of VSC in the simplest possible scenario, hence we only consider a single nuclear DOF in the reaction coordinate.

In the present work, we only consider adiabatic reaction (by only including the ground electronic state in our theoretical description). This is valid for the molecular system we studied, because the higher excited electronic states are well separated from the ground state by at least 2.7 eV (see Supplementary Fig. 3 in the Supplementary Information). Note that the kinetic effect could also manifest into a non-adiabatic reaction that involves multiple electronic states, such as recent theoretical work by Nitzan and coworkers [J. Chem. Phys. 150, 174122 (2019)] as well as Yuen-Zhou and coworkers [arXiv:2011.08445, and Nat. Comm. 10, 4685 (2019)] that showed the modification of ground state Electron Transfer reaction. If we want to study the VSC regime of non-adiabatic reaction while still treating photonic DOF through the quasi-classical description, then we need the non-adiabatic rate constant beyond simple Marcus theory (which includes non-Markovian dynamics of the photonic coordinate). Examples of these theories can be found in the recent work of Geva and co-workers through the linearized path-integral rate expressions that explicitly include these effects [J. Phys. Chem. A, 120, 2976 (2016), J. Chem. Phys. 144, 244105 (2016), J. Chem. Phys. 148, 102304 (2018)]. On the other hand, we find that through a ring polymer path-integral treatment of the cavity mode, we can also accurately simulate the reaction rate in both

the non-adiabatic and the adiabatic regimes of polariton mediated electron transfer [Chowdhury, Mandal and Huo, DOI:10.26434/chemrxiv.13082216.v1]. This will open up new doors for investigating the non-adiabatic reactions under the VSC coupling regime.

Overall, this present work provides a new perspective and a new direction towards resolving open questions in VSC, which may finally provide us with the understanding of recent experimental observations. **No change has been made.**

Comment 5. Which electronic structure methods could be used for a first principle description of these effects? Are ab-initio methods for strong light-matter systems, such as in Phys. Rev. X 10, 041043 2020, Chem. Phys. 2018,509, 55-65, or Phys. Rev. Lett. 121, 113002 (2018) sufficient, or are there missing parts that need to be developed?

Response: In the present approach, we rely on using molecular ground adiabatic potential energy surface and the ground permanent dipole as well as their derivatives with respect to the nuclear coordinate, to describe the light-matter system. Any approach (essentially any ab-initio electronic structure method) that provides these quantities are well suited for computing the transmission coefficient with the flux-side formalism. This is because the photonic coordinate is treated classically and is evolved over the CBO surface (such Fig. 3b-d).

On the other hand, the new polariton ab-initio methods mentioned by the reviewer for strong light-matter systems, aim to resolve electronic polaritonic eigenstates and polaritonic potential energy surfaces, such that the photonic degrees of freedom are treated quantum mechanically. To the best of our knowledge, these approaches can not directly provide the *vibrational polaritonic state* information (through the procedure outlined in Section 5.b of Supplementary Information that involves phonon creation, annihilation operators in the Hamiltonian), hence they are not directly applicable in the VSC chemistry investigations.

No additional changes are made.

Reviewer #2:

Comment (Remarks to the Author): The manuscript by Li, Mandal and Huo discusses a possible theoretical explanation for the observed phenomenon, that coupling to an optical cavity suppresses chemical reaction rates. The approach introduces a new idea, focusing on the dynamics of the reaction, and how the cavity degree of freedom can change the transmission coefficient. This is illustrated with a combination of analytical and numerical calculations, studying in detail a simple model of reaction kinetics.

I believe this is a significant paper. Prior to this work, there was no viable theoretical explanation of several features of the experimentally observed behavior. In particular, it was hard to explain why the reaction rate would show a resonant enhancement at a particular cavity frequency. The current manuscript provides a possible explanation of this observation. There are still some questions open. In particular, as noted by the authors, the question of how this idea generalizes for multiple molecules coupled to the same cavity. These are important questions, but I believe it is appropriate to publish the current manuscript without these. The manuscript presents and describes clearly a new idea that may be significant in understanding the effect of cavities on chemical reactions.

There are some presentational issues I would suggest the authors consider. These mostly arise from the fact this field is one involving both physicists and chemists. The manuscript as it stands is perhaps more targeted to an audience of chemists. Given the potential broad audience of this paper, it would be helpful to make sure it was accessible also to physicists.

Response: We thank the reviewer for the encouraging comments as well as pointing out the presentational issues. We will follow the suggestions from the reviewer to make the manuscript suitable for a broader audience.

Comment 1: Main text page 3. There are references in this section to results "presented in the main text". This seems an odd phrase to use in the main text; perhaps this could be replaced by a more specific reference, e.g. "the $\hbar\Omega_R$ presented in Fig. 2"

Response: We thank the reviewer for pointing this out. As the exact values of Ω_R only showed up in Fig. 2, the phrase "presented in Fig. 2" is indeed more appropriate to use in this context.

Change: We changed "The $\hbar\Omega_R$ presented in the main text" to "The $\hbar\Omega_R$ presented in Fig. 2".

Comment 2a: Main text Eq. 4, and supplemental Eq. 104. There are a few issues here: Firstly, I am unclear what is meant by the time derivative of $R\ddagger$ in the denominator. $R\ddagger$ is defined as being the location of the dividing surface, so does not change with time. Should this be the time derivative of R instead?

Response: The time derivative of $R\ddagger$ at $t = 0$ represents the initial velocity of the nuclei on the dividing surface.

Change: To clarify this, we have explicitly stated the meaning of the time derivative of $R\ddagger$ in the main text after Eq. 4, with “Further, $\dot{R}_{\ddagger}(0)$ represents the initial velocity of the nuclei on the dividing surface.” Similar changes were also made to the paragraph following Supplementary Eq. 104.

Comment 2b: Secondly, this formula is not necessarily familiar to physicists. It would be helpful to explain its meaning more clearly, in addition to referencing relevant papers.

Response: We thank the reviewer for pointing this out. The flux-side formalism is indeed a more technical result that is well-known in the chemistry community. In fact, it can be derived by assuming linear response theory, which gives the rate constant expression as a flux-side correlation function, for example, in a standard textbook [Introduction to Modern Statistical Mechanics, D. Chandler, Oxford University Press, 1987], page 242, section “8.3 Application: Chemical Kinetics”.

Change: We cite the above standard statistical mechanics textbook after we introduce this equation (Page 3): “It can be numerically calculated from the flux-side correlation function formalism [new Ref] as follows”

Comment 2c: Thirdly (related to the point above), I am somewhat confused why it is necessary to specify that the ensemble average is on the dividing surface when the flux contains a delta function that limits one to that dividing surface.

Response: We thank the reviewer for pointing this out. The delta function in the flux operator indeed implies the constraint, which renders the double dagger of the ensemble average redundant.

Change: The double daggers have been removed from both the numerator and denominator in Eq. 4 and the Supplementary Eq. 104.

Comment 3: Main text page 4, around Eq. 5. While I agree that this equation is the minimal result of this calculation, I think the existence of this simple form is important enough that it warrants a bit more explanation in the results, rather than having all details in the methods. I might suggest in particular Eq. 8 from the methods, and some of the following discussion. This would help make clear where this formula comes from --- i.e. that it comes from the form of a vibrational frequency Ω_{-}

Response: We thank the reviewer for this suggestion. We have added new discussions before the original equation Eq. 5 (now Eq. 6 in the modified manuscript). In particular we have introduced the Eq. 8 (now Eq. 9) in the main-text before introducing the expression for the transmission coefficient.

Change: On page 3 on the right column, we added one new expression (now equation 5) and several sentences starting with “The total rate constant k in the GH theory can be obtained...”

Comment 4: Main text, Page 5, there is a reference to Fig. 2e presenting a typical "dynamical caging" regime; I think this means Fig. 2f.

Response: We thank the reviewer for pointing out the typo. We have corrected it.

Change: The main text, Page 4 on the present manuscript, the first line of the right column, “Fig. 2e presents a typical “dynamical caging” regime of the GH theory”. “Fig. 2e” has been changed to “Fig. 2f”.

Comment 5: Supplement, Figure 1 caption. This caption appears to use the phrase "Langevin limit" (three times) to refer to the Markovian limit.

Response: We thank the reviewer for pointing this out. All three occurrences should be “Markovian limit”, instead of “Langevin limit”. We have corrected all of them.

Change: All there “Langevin limit” in Supplementary Fig. 1 caption has been changed to “Markovian limit”.

Comment 6: Supplement, inline equation between Eq. 92 and 93. The force on the nuclear coordinate seems to be missing a term from the derivative of $E(R)$.

Response: We appreciate the reviewer for finding out this typo. Indeed, there should be a derivative of $E(R)$. We have added the term.

Change: The missing term $E'(R)$ has been added to the inline equation between Supplementary Eq. 92 and Eq. 93.

Reviewer #3:

Comment: In "Resonant Theory of Vibrational Strong Couplings in Polariton Chemistry," Li and co-workers address the contemporary problem of vibrational-strong-coupling induced modification of adiabatic chemical kinetics. They expand upon previous cavity Born-Oppenheimer based approaches by considering VSC as having dynamical implications instead of only equilibrium ones. Within the frame of Grote-Hynes transition-state theory, the authors find that the rate constant does have a frequency dependence, which has not been explored in previous studies. They confirm their findings by performing a numerical simulation within the Shin-Metiu model. These analyses allow them to conclude that VSC produces an effect akin to the dynamical solvent caging, which modifies reaction rates in solution compared to in gas phase. Moreover, the authors explain the failure of previous approaches in terms of the resonance effects having a different nature from those that describe the absorption at polaritonic frequencies. This is a remarkable finding that has evaded several attempts to explain Ebbesen and collaborators' results and is a sound step forward in understanding the observed effects.

Response: We thank the reviewer for the encouraging comments.

Comment 1: However, although this work shines a light on one of the main mysteries surrounding thermally activated reactions in infrared cavities, namely the frequency-dependent effects, it still falls short at addressing the collective nature of the phenomena, which has not only been confirmed experimentally but also determined to be an essential feature of the observations.

Response: We appreciate the reviewer's concern and agree with the reviewer in that the present work is strictly restricted to a single molecule coupled to a single-mode in a cavity. As we have already clearly pointed out in our original draft, we do not have a theoretical explanation for the observed collective effect. Moreover, **the focus of the current paper is not on explaining the collective effect**, but rather on the resonant behavior of VSC. To the best of our knowledge, all of the previous theoretical work on adiabatic chemical reactions based on TST theory failed to explain any resonant effect (for a single molecule or N molecules coupled to the cavity). Hence, the intellectual merit of the current draft is that it successfully explains the resonant effect, even though with just a single molecule coupled to the cavity. **No change has been made.**

Comment 2: The authors claim that their "theoretical hypothesis provides a plausible explanation to the observed resonant effects of the electronically adiabatic ground-state reactions coupled to an optical cavity" without discussing its obvious shortcomings: the transition dipole moment they assigned to the unstable mode is unrealistically high, which is an indication of the need of collective effects.

Response: We respectfully disagree with the reviewer that the dipole we used here is unrealistically high. All of these dipoles (as well as all parameters) are directly derived

from a realistic electron-proton transfer model (the Shin-Meitu Model) that has been well studied. On the other hand, we agree with the reviewer that the strength of the molecule-cavity in our current model would be much stronger than the realistic coupling strength in the VSC experiments, because we do not include many molecules (hence no collective coupling effects explicitly in our model).

Change: On page 2, at the end of the first paragraph of the result section, we have added “Note that the molecule-cavity coupling strength per molecule used in this work would be much stronger than the realistic coupling strength in the VSC experiments [ref] that includes many molecules. On the other hand, the Rabi splitting (from the IR spectrum) of the current work is within the range of the recent VSC experiments [ref]. This is because in these VSC experiments, the collective coupling strength is scaled up by \sqrt{N} .”

Comment 3. Nonetheless, the theoretical basis upon which their results hold is applicable only when a single molecule (or possibly a few) couples to the cavity; in the collective regime, the arguments presented in refs 17, 19 and Chem. Phys. 535, 110767 (2020) (which the submission does not cite) would also nullify the findings of the manuscript at hand.

Response: We agree with the reviewer that, as shown in Ref. 17, 19, and 27 (Chem. Phys. 535, 110767, 2020), this effect in the collective regime remains an open question. (Note that we have already cited Chem. Phys. 535, 110767, 2020 in our original version of the manuscript). However, we respectfully disagree with the reviewer in that we don't think that the findings of Ref. 17, 19, or 27 necessarily nullify the finding of the present manuscript. Despite the thorough theoretical work done in these references, they also fall short in considering terms that might ultimately lead to a collective effect when extending our present work. The present formalism, which is computing the transmission coefficient from through the derivation of generalized Langevin equation, when extended to include the cavity and multimolecular dissipation or when coupling a cavity to bath modes (which couple to the reaction coordinate) and the interplay of cavity induced caging and existing solvent cage might provide a different answer than what is shown in the referred papers. Therefore, we believe that the present work provides the first step towards the ultimate theoretical understanding of the VSC experiments. Obviously, these are the subject of our present and future research. We added a new paragraph listing the concerns of the reviewer.

Changes: On page 5 of the current manuscript we added a new paragraph clarifying the limitation of this work:

“We want to remind the reader that the present work is limited to a single molecule coupled to a single radiation mode, whereas the experimentally observed frequency-dependent modification of the chemical kinetics in the collective coupling regime. It was suggested in [Ref] that the resonant effects will disappear under the $N \rightarrow \infty$ limit, where N is the number of molecules coupled to the cavity. However, we believe that the effect we have seen will be present under the few N limit (in the current paper, $N=1$). Whether our current theory can also be extended to the collective regime

remains an open question. The present formalism can be extended to include cavity losses and their impact on the caging effect. The VSC experiments often use low-Q cavities, where the cavity loss should also be explicitly included. Interestingly, these far-field modes which are responsible for the cavity loss can be modeled as dissipative modes coupled to the quantized modes of cavity, providing an additional dissipative environment for the hybrid system. We hypothesize that the frequency dependence as well as the collective phenomenon would emerge when cavity loss is also included. [Ref] ”

Comment 4: Discourse-wise, I am also uncomfortable with the authors' language referring to their finding as a resonant effect instead of a frequency-dependent one (in fact, a relatively low frequency one when compared to the IR cavity modes used in the Ebbesen experiments). Although the authors emphasize that the tuning role is different, I still think it might be misleading given the general understanding of this term in the community. Moreover, they found maximal effect without any resonance condition being strictly met. Thus, I suggest using the language of "frequency-dependent" highlights the true nature of the improvement of their approach over the previous ones.

Response: We want to remind the reviewer that the resonant condition for observing a minimal rate constant is clearly presented in Eq. 7 (of the current draft), and we are afraid that the reviewer is incorrect to suggest that “they found maximal effect without any resonance condition being strictly met”. We also respectfully disagree with the reviewer on the comment “a relatively low frequency one when compared to the IR cavity modes used in the Ebbesen experiments”. Note that in the weak coupling regimes ($\eta < 0.1$, which is the green and blue curves in Fig. 3A of the manuscript), the resonance frequencies are within the range of 90-100 meV ($700\text{-}800\text{ cm}^{-1}$), which is similar to what have been observed in experiments [Science, 363, 615 (2019)] Only when we reaching to the Ultra-strong coupling regimes ($\eta > 0.1$, which is the red curve in Fig. 3A of the manuscript) do we see a significant redshift of the resonant condition, reaching a resonant frequency of only 10 meV. We want to remind the reviewer that the USC coupling regime has not been achieved in the current experimental investigations of VSC. Our theory gives such a prediction that (at least when a single molecule is coupled to the single cavity mode) a significant red-shift will be observed in the USC regime of VSC.

On the other hand, we do agree with the reviewer that the “frequency-dependent modification” is a more appropriate term for the theoretical observation in our work. We have made the following changes to clarify our use of the term “resonant effect”.

Changes:

- In the abstract: “...plausible explanation of the resonant modification...” to “...**plausible explanation of the photon frequency dependent modification...**”
- On page 1, right column: Added a new sentence clarifying the term “resonance effect”: “**We note that in this work we refer to the photon frequency dependent**

modification of the ground state kinetics as the resonance effect.”

Comment 5: Finally, the authors give a hand-wavy explanation for the coincidental observation of resonant effects by claiming "however, it is possible for a given molecular system which has[sic] $\omega_0 \approx \omega_b$ ". For the sake of completeness, a reader might find useful a table that compares typical values of both frequencies for common S_N2 reactants. As explained above, I find this work relevant and timely, and I will gladly support its publication in Nature Communications if the authors adequately address my concerns.

Response: We thank the reviewer for encouraging us to connect with real chemical systems that could potentially satisfy this condition. However, we are afraid that preparing a table that documents available S_N2 reactions is beyond the scope of the draft. Nevertheless, we would like to point the reviewer to a well known theoretical work by Angela Merkel and co-workers on an S_N2 reaction, which indeed satisfies this condition.

Change: On page 5 of the current manuscript, left column, we have added “**For example, in a theoretical work (at the level of MP2 perturbation theory) by Merkel and co-workers^{~\cite{Merkel1988}}, a well-studied S_N2 reaction (CH₃F + H⁻ → CH₄ + F⁻) has a $\omega_b=975.5$ cm⁻¹, which is close to one ground state vibrational frequency $\omega_0=978.7$ cm⁻¹. In fact, this reaction could be also an ideal one subject to future investigations of VSC modifications of reactivities. “**

REVIEWERS' COMMENTS

Reviewer #1 (Remarks to the Author):

The authors have sufficiently addressed my comments and I can recommend the paper for publication.

Reviewer #2 (Remarks to the Author):

The revised manuscript addresses all the presentational issues I raised previously. For the reasons stated in my previous report I fully support publication. On rereading the manuscript I noted two minor typos in the supplement:

Page 4, the word Hessian is repeated twice

Page 19, there is an unresolved reference

Reviewer #3 (Remarks to the Author):

After reviewing the author's responses to all referees' concerns, I would like to clarify a couple of things.

First, I stand by my claim that the maximal effect takes place without a strict resonant condition being met. The suggested –and discussed– resonance is between ω_c and ω_b (as opposed to the expected resonance between ω_c and ω_0 , which is the experimentally observed one). Since the light-matter coupling needs to be non-zero, per Eq. 7 of the corrected manuscript, the two frequencies ω_c and ω_b never have the same numerical value. Regardless, I am satisfied with the corrections the authors made to the document regarding its language on resonances.

The second thing I would like to discuss is the connection to experiments. While I appreciate the mention of the system studied by Merkel and coworkers, I think the current paper should also include a counterexample to the statements in the experimental literature, namely, where ω_b and ω_0 are very different from each other.

Given the two items above, I believe that the current title is misleading; it would be better if the title were "Frequency-dependent Vibrational Strong Couplings in Polariton Chemistry".

If the changes above are made, I am ready to support the publication of their work in Nature Communications.

Reviewer #1 (Remarks to the Author):

The authors have sufficiently addressed my comments and I can recommend the paper for publication.

Response: We thank the reviewer for recommending this work for publication.

Reviewer #2 (Remarks to the Author):

The revised manuscript addresses all the presentational issues I raised previously. For the reasons stated in my previous report I fully support publication. On rereading the manuscript I noted two minor typos in the supplement:

Page 4, the word Hessian is repeated twice

Page 19, there is an unresolved reference

Response and Changes: We thank the reviewer for supporting the publication of the manuscript and for pointing out the typos. We have deleted the repeated “Hessian” (in Supplementary Page 4) and have fixed the issue with the reference (in the Supplementary Page 19).

Reviewer #3 (Remarks to the Author):

After reviewing the author’s responses to all referees’ concerns, I would like to clarify a couple of things.

First, I stand by my claim that the maximal effect takes place without a strict resonant condition being met. The suggested –and discussed– resonance is between ω_c and ω_b (as opposed to the expected resonance between ω_c and ω_0 , which is the experimentally observed one). Since the light-matter coupling needs to be non-zero, per Eq. 7 of the corrected manuscript, the two frequencies ω_c and ω_b never have the same numerical value. Regardless, I am satisfied with the corrections the authors made to the document regarding its language on resonances.

Response and Changes: We agree with the reviewer that the condition for resonant will not be $\omega_c = \omega_b$, that is why we express the condition as $\omega_c \approx \omega_b$ in the main text. We are glad to learn that our modification in the language “on resonance” satisfied the reviewer.

The second thing I would like to discuss is the connection to experiments. While I appreciate the mention of the system studied by Merkel and coworkers, I think the current paper should also include a counterexample to the statements in the experimental literature, namely, where ω_b and ω_0 are very different from each other.

Response and Changes: We have also added a counter example in page 5 of the present manuscript starting with “On the other hand, there are also cases where....”.

Given the two items above, I believe that the current title is misleading; it would be better if the title were “Frequency-dependent Vibrational Strong Couplings in Polariton Chemistry”. If the changes above are made, I am ready to support the publication of their work in Nature Communications.

Response and Changes: We have modified the title of our manuscript to “Frequency-dependent Vibrational Strong Couplings in Polariton Chemistry” as suggested by the reviewer.